# Investigating soil moisture-climate interactions with prescribed soil moisture experiments: an assessment with the Community Earth System Model (version 1.2)

Mathias Hauser[1], René Orth[1], and Sonia I. Seneviratne[1]

[1]Institute for Atmospheric and Climate Science, ETH Zurich, Zurich, Switzerland

*Correspondence to:* Mathias Hauser (mathias.hauser@env.ethz.ch)

**Abstract.** Land surface hydrology is an important control of surface weather and climate. A valuable technique to investigate this link is the prescription of soil moisture in land surface models, which leads to a decoupling of the atmosphere and land processes. Diverse approaches to prescribe soil moisture, as well as different prescribed soil moisture conditions have been used in previous studies. Here, we compare and assess four methodologies to prescribe soil moisture and investigate the impact of two different estimates of the climatological seasonal cycle used to prescribe soil moisture. Our analysis shows that, though in appearance similar, the different approaches require substantially different long-term moisture inputs and lead to different temperature signals. The smallest influence on temperature and the water balance is found when prescribing the median seasonal cycle of deep soil liquid water, whereas the strongest signal is found when prescribing soil liquid and soil ice using the mean seasonal cycle. These results indicate that induced net water-balance perturbations in experiments investigating soil moisture-climate coupling are important contributors to the climate response, in addition to the intended impact of the decoupling. These results help to guide the set up of future experiments prescribing soil moisture, as for instance planned within the "Land Surface, Snow and Soil Moisture Model Intercomparison Project" (LS3MIP).

## 1 Introduction

The interplay between the land surface and the atmosphere can induce or modulate anomalies in temperature (Hirschi et al., 2011; Whan et al., 2015) and precipitation (e.g. Guillod et al., 2015). Soil moisture (SM) is a key quantity in this context (Seneviratne et al., 2010). The complex role of SM in land-atmosphere dynamics can be investigated with General Circulation Models (GCMs). Typically in this context, land state variables are set – prescribed – to predefined target values in GCM simulations. Such experiments are performed since decades (e.g. Shukla and Mintz, 1982). Prescribing land state variables suppresses interactions between the land and the atmosphere and can hence be used to infer the role of land-atmosphere interactions for the climate.

The Global Land Atmosphere Coupling Experiment (GLACE, Koster et al., 2004, 2006) was the first major multi-model effort to comprehensively analyse the impact of SM on several atmospheric variables in the context of present climate. In multi-model simulations of a particular northern hemisphere summer, regions of coupling between precipitation and evaporation were identified. While some regions emerged as multi-model 'hot spots', the experiment revealed a large inter-model spread in the

land-atmosphere coupling strength, pinpointing to different sensitivities of the models with respect to the link between SM and evapotranspiration, and the link between evapotranspiration and precipitation (Guo et al., 2006).

More recently, the role of SM-climate feedbacks in climate change projections has been investigated in the multi-model project Global Land-Atmosphere Coupling Experiment of the Coupled Model Intercomparison Project, Phase 5 (GLACE-CMIP5, Seneviratne et al., 2013). In GLACE-CMIP5, an ensemble of GCMs performed two distinct experiments for the period 1950 to 2100 to assess the role of inter-annual SM variability, and of SM trends for climate change simulations. The removal of both, interannual SM variability and the long-term SM trend by prescribing the mean seasonal cycle from 1971 to 2000 ('experiment A', Seneviratne et al., 2013), leads to large decreases in temperature extremes as well as effects on precipitation extremes (Seneviratne et al., 2013; Lorenz et al., 2016; Vogel et al., 2017). In another experiment ('experiment B') the 30-year running mean of the reference experiment is prescribed to preserve long-term SM trends. Projected SM drying trends were found to be accompanied by a further increase of temperature extremes. However, the simulated SM trends were strongly model-dependent.

In the context of the upcoming CMIP6 modelling cycle, the Land Surface, Snow and Soil Moisture Model Intercomparison Project (LS3MIP, van den Hurk et al., 2016) plans a variety of experiments to quantify and compare the role of multiple land state variables in climate change simulations. Particularly, the Land Feedback Model Intercomparison Project (LFMIP) within LS3MIP plans experiments similar to the GLACE-CMIP5 project which aim to quantify the role of land-atmosphere feedbacks at the climate time scale. In contrast to the GLACE-CMIP5 experiments, simulations will be run with an interactive ocean.

Additionally to the above-mentioned GLACE-type experiments, a large number of studies analysed the influence of SM on the atmosphere from multiple perspectives (e.g. Koster et al., 2000; Douville et al., 2001; Reale and Dirmeyer, 2002; Douville, 2003; Seneviratne et al., 2006b; Rowell and Jones, 2006; Vautard et al., 2007; Fischer et al., 2007a, b; Conil et al., 2007; Jaeger and Seneviratne, 2011; Lorenz et al., 2012; Hauser et al., 2016; Douville et al., 2016; Orth and Seneviratne, 2017, early online release). The different goals, and also the different employed land surface models in these studies motivated, and necessitated different techniques to prescribe SM. They include the prescription of (1) all land state variables, (2) only SM at all soil depths, (3) SM in subsurface soil layers only, (4) nudging SM values, and (5) restricting the SM prescription to certain regions. In addition, the prescribed SM values vary widely between studies. Some use the plant wilting point and the field capacity to simulate extreme dry and wet conditions, respectively. Others use simulated SM from a particular year, a climatological seasonal cycle, or a smoothed seasonal cycle. Furthermore, the SM climatology can be estimated (calculated) in different ways: either using the *mean* (as done in e.g. Seneviratne et al., 2013) or the *median* (as done in Orth and Seneviratne, 2017). A third difference between the SM-prescription methodologies is the temporal resolution of the SM target dataset – they comprise instantaneous, daily, and interpolated monthly data.

Similarly to prescribing sea surface temperatures in GCMs, which does not allow for conservation of the energy balance, modelling experiments prescribing SM infringe the water balance of the land model. However, water is only added or removed by the prescription algorithm within the soil and not in the atmosphere or at the land-atmosphere interface. Thus, and because such experiments analyse only the atmospheric response, the perturbation of the soil water balance is 'deemed acceptable' (Koster et al., 2006). Still, prescribing SM induces artificial sources and sinks of water in the model. To our knowledge a

quantification of this water-balance disturbance and its impact is currently lacking. In particular, the distinct effects of different existing methodologies on these water imbalances and their impact have not been systematically compared so far. This is an important gap because it is possible that they could lead to methodologically-induced discrepancies between studies.

In the present article, we analyse differences in SM-prescribing set-ups that aim to remove the inter-annual variability while conserving the seasonal cycle of SM to assess its impact on surface climate. In this context, we focus on methodologies which are relevant for the LS3MIP experiment such that our conclusions can contribute to the final implementation of its experimental design.

## 2   Model description

In this section we first introduce the employed GCM and the corresponding land surface scheme. Thereafter, we describe the different tested approaches to prescribe SM. Finally, we provide an overview of the conducted experiments.

In this study, we use the Community Earth System Model (CESM, Hurrell et al., 2013, version 1.2). This is a fully coupled Earth System Model, combining separate modules for the atmosphere, the ocean, and the land. Land surface processes and their coupling to the atmosphere are simulated by the Community Land Model, version 4.0 (CLM4, Lawrence et al., 2011). CLM4 is a third-generation land surface model (Sellers et al., 1997; Pitman, 2003), incorporating the hydrological cycle (see below), land surface energy fluxes, a variety of land surface types (wetlands, glacier, vegetated, etc.) and up to 15 generic plant types ('plant functional types'), among others.

### 2.1   Short overview of hydrology in the Community Land Model

Water in CLM4 is stored in four reservoirs: on the canopy, as snow, as groundwater, and in the soil. The soil is divided into 15 vertical layers with exponentially increasing thickness from top to bottom. However, only the ten first layers are hydrologically active and extend to a depth of 3.8 m (the last five layers act only as thermal sink/ source). Water reaching the soil surface through precipitation and stemflow is partitioned into surface runoff and infiltration, i.e. water entering the uppermost soil layer. Water is removed from the soil by subsurface runoff (drainage) and canopy transpiration through root extraction. The water flux within the soil is governed by Darcy's Law. The corresponding hydraulic properties are a function of soil water content and texture. Water can occur in liquid and solid states, which will be referred to as LIQ and ICE for the remainder of this study. A comprehensive description of CLM4 can be found in Oleson et al. (2010).

### 2.2   Prescription of soil moisture in the Community Land Model

The aim of SM prescription is to control the soil's water content, i.e. to force it to a predefined target value (e.g. a climatological seasonal cycle, the plant wilting point or others), irrespective of the actual conditions in the soil. As this is not possible with the default model version, we extend the original model code of CLM4 with a module (see Section 4.1) that reads the target value from a previously prepared file and overwrites the actual value in the model after each time step. The goal of this study

is to assess and compare various approaches of prescribing SM. The tested techniques comprise established as well as novel methods as listed in Table 1 and Figure 1.

In previous studies (Koster et al., 2006; Lorenz et al., 2012; Seneviratne et al., 2013) SM in CLM was prescribed by setting LIQ and ICE individually to the predefined values at each time step (Figure 1a). This technique will be referred to as PRES_LIQ+ICE. A second technique, named PRES_FRAC (Figure 1b) also prescribes LIQ and ICE, but lets the land surface model interactively compute the fraction of LIQ (e.g. applied in Douville et al., 2016). Hence, the model has an additional degree of freedom compared to PRES_LIQ+ICE.

Furthermore, we propose an alternative approach where SM is only prescribed when the soil temperature is above $0$ °C (PRES_LIQ). If the soil is frozen, LIQ and ICE are both computed interactively. The climatological total SM (i.e. LIQ + ICE) is converted into LIQ for the prescription. The important characteristic of this new algorithm is that it never artificially adds ICE (see Section 3.2.2). Although (supercooled) LIQ and ICE can coexist in CLM4, we leave the soil hydrology entirely interactive below the freezing temperature. In detail the algorithm works as follows: LIQ is prescribed starting from the uppermost soil level, and then further down until either the soil bottom is reached, or until a layer with soil temperature at or below $0$ °C is found (Figure 1c). This follows the methodology employed in the (optional) irrigation module of CLM (Oleson et al., 2013).

Following an approach presented in Douville (2003), and also used in Koster et al. (2006), we additionally test a similar methodology as in PRES_LIQ, but without prescribing the topmost soil layer, hereafter named PRES_LIQ_DEEP (Figure 1d). Whereas in the other prescription approach the land-atmosphere coupling is entirely removed, this allows for a limited feedback between the soil and the atmosphere. Even though the topmost layer is only $1.8$ cm thick, it controls bare-soil evaporation, which forms a significant part of the total evapotranspiration. Additionally, SM in the topmost layer – in contrast to the deep(er) soil layers – may not be well predictable as it does not have its considerable inertia and memory (Koster and Suarez, 2001; Seneviratne et al., 2006a; Orth and Seneviratne, 2012).

For all four methods the hydrology in CLM4 is still active – SM is removed by root extraction and drainage and added by infiltration. However, at the end of each time step, this interactively calculated SM is overwritten and set to the target value. We record the difference of the interactively computed SM and its target value as the water-balance perturbation. If it is positive, the algorithm has artificially 'added SM', while it has 'removed SM' if the difference is negative.

Finally, we have to choose the time resolution of the SM dataset from at least four possibilities: (1) monthly data with linear interpolation to daily mean values, (2) daily mean values, (3) daily mean values with linear interpolation to every model time step, and (4) instantaneous values at every model time step. In this study we use daily mean values as linearly-interpolated monthly values can be too coarse (see below).

## 2.3 Overview of the experiments

All simulations (Table 1) are conducted with CESM. As reference simulation we perform a fully coupled simulation from 1950 to 2099 (hereafter called REF), combining the historical forcing and the Representative Concentration Pathway 8.5 scenario (RCP8.5, Meinshausen et al., 2011). The daily SM output from REF between 1971 and 2000 is used to calculate the mean and median climatology at every grid point, soil level, and day of the year for LIQ and ICE individually.

We perform seven simulations with prescribed SM that differ in the method to prescribe SM (Section 2.2 and Figure 1), and the target SM climatology. In the simulations with prescribed SM, we also prescribe sea surface temperatures (SSTs) and sea ice from REF to suppress impacts from changed SSTs in response to the prescribed SM (as done in GLACE-CMIP5, Seneviratne et al., 2013). The first two simulations (PRES_LIQ_MEAN and PRES_LIQ_MEDIAN) use the new SM prescription scheme

described above with mean and median climatologies, respectively. The third simulation, PRES_LIQ_DEEP_MEDIAN, also uses the new prescription scheme but leaves the first layer interactive. In the fourth and fifth simulation (PRES_LIQ+ICE_MEAN and PRES_LIQ+ICE_MEDIAN), we prescribe LIQ and ICE and also compare mean and median SM climatology. Finally, simulations six and seven also prescribe LIQ and ICE, but calculate the respective fractions interactively (PRES_FRAC_MEAN and PRES_FRAC_MEDIAN). In our analysis we concentrate on the simulations that do *not* prescribe ICE because both tech-

niques that do so lead to large, unrealistic surface temperature and ground heat flux anomalies (see Section 3.2.2). The variety of soil moisture prescription approaches considered here makes this study a valuable basis for the final planning of the soil moisture prescription methodology for the LS3MIP simulations.

## 3   Results and discussion

### 3.1   Soil moisture climatology

#### 3.1.1   Mean vs. median soil moisture

The daily mean and median SM climatologies only differ if the inter-annual SM values are not symmetrically distributed. As an example, Figure 2a shows the evolution of SM throughout the year in the topmost $10\,\mathrm{cm}$ of the soil for a location in India. This grid point shows a distinct seasonal cycle with a dry period from February to May and high soil moisture values during the rest of the year. In the dry season the median is generally smaller than the mean, with large rainfall events leading to outliers on

the wet end of the distribution. For example on the 5[th] of April, the difference is $-2.3\,\mathrm{mm}$, or $-14.0\,\%$ (Figure 2b). During the wet period the median is usually larger than the mean, and it is dry years that lead to the asymmetry. However, the difference between median and mean are generally smaller than during the dry period; e.g. on the 21[st] of December it is $1.0\,\mathrm{mm}$, or $3.8\,\%$ (Figure 2c). There are many processes that contribute to non-symmetric SM distributions: the positively skewed distribution of precipitation, the upper and lower bound in the water holding capacity of the soil (between the wilting point and saturation), as

well as the strong nonlinear function of water flow (hydraulic conductivity) within the soil with respect to the SM state (Laio et al., 2001).

In Figure 2d and e, we show the relative difference between the mean and median SM for two depth intervals. We thereby focus on the three hottest consecutive months of the year, as we expect SM differences in these months to have the largest temperature impact. The hottest months of the year are determined from REF. On land in the mid- and high latitudes these three

hottest consecutive months generally correspond to summer, i.e. June to August in the Northern Hemisphere and December to February in the Southern Hemisphere (Figure S1). The largest relative differences between the mean and median are found in the uppermost $10\,\mathrm{cm}$ of the soil (Figure 2d). Regions for which the median is drier than the mean include Australia, North

Africa, the Mediterranean, and Western America, while it is wetter in Central Africa, Central Europe, Western Asia and Central North America. As for the example grid point in India, negative differences are generally stronger than positive differences. In contrast to these large relative SM differences in the top 10 cm, the relative differences are generally below 2 % in depths between 10 cm and 100 cm (Figure 2e), and from 100 cm to 380 cm (not shown). The absolute differences, however, are higher for deeper soil levels, as these are thicker. A difference between the mean and median climatologies in the topmost 10 cm of the soil is not only a feature of CESM but it is also evident in other models participating in GLACE-CMIP5 (Figure S2).

### 3.1.2 Daily vs. interpolated monthly soil moisture

In this study we prescribe daily SM values whereas some previous studies used daily values obtained from a linear interpolation of monthly means (e.g. some simulations in the GLACE-CMIP5 experiment, Seneviratne et al., 2013). True daily and interpolated monthly SM values can differ in regions with a short sharp peak in the seasonal cycle, as exemplified for a grid point in Central Africa (Figure 3a). It shows true daily values (blue line) and the corresponding monthly means (blue dots). The orange line illustrates daily values linearly interpolated from the monthly mean values, where these monthly values were assumed to occur in the middle of each month. While true daily and interpolated monthly values match closely for most of the year, the latter does not entirely capture the summer minimum. In addition, the monthly means derived from the interpolation (orange dots) are not equal to the true monthly means derived from the daily time series. In contrast, the annual mean of the daily and monthly interpolated values are equal.

We show the median absolute differences of the warm season months between true daily and interpolated SM values in Figure 3b and c. While the difference is generally smaller than between mean and median SM climatologies, it is comparable in some regions, e.g. the Sahel, Southern Africa, and Australia (Figure 2b). For the depth intervals 10 cm to 100 cm, and 100 cm to 380 cm (not shown), the relative difference is generally below 2 %. In contrast to the difference between the mean and median SM climatologies, positive and negative deviations between daily and interpolated monthly SM climatologies compensate when integrated over time. This analysis shows that other methodological differences apart from using mean or median seasonal cycle may (regionally) cause important implications.

## 3.2 Temperature response

### 3.2.1 Prescribing soil liquid water only

In this section we investigate the influence of the newly developed SM prescription methodologies on surface air temperature. Figure 4a to c show the climatological temperature between 1971 and 2000 for each methodology compared to REF. In all three simulations the mean land temperature is lower than in REF. The largest difference is found for PRES_LIQ_MEAN which has negative temperature anomalies for almost all land grid points. PRES_LIQ_MEDIAN has smaller temperature anomalies than PRES_LIQ_MEAN, corresponding to the regions with smaller climatological SM when comparing the median to the mean (Figure 2). For PRES_LIQ_DEEP_MEDIAN we obtain the smallest anomalies. We find similar results when comparing the experiments to REF for the time period 2070 to 2099 (Figure S3a to c). Thus, the global land warming between 1971 to 2000

and 2070 to 2099 is only slightly larger in REF than the experiments. This is in line with earlier findings (Seneviratne et al., 2013), although experiments in this study are at the lower end of the range of the individual GLACE-CMIP5 models.

In addition to changes in annual mean temperature in response to prescribed SM, we also investigate corresponding changes in annual maximum daily maximum temperature (TXx), shown in Figure 4d to f. In most regions the TXx differences are larger than the annual mean differences. This stronger impact of SM changes on extremes versus mean temperatures is a well-known characteristic of land-atmosphere coupling (e.g. Seneviratne et al., 2010, 2013). TXx in PRES_LIQ_MEDIAN are cooled by more than 2 °C by the SM prescription for Australia, South Africa, India and Brazil. The results for PRES_LIQ_DEEP_ MEDIAN are similar to PRES_LIQ_MEDIAN, except in South Australia and Northern High Latitudes. These results are in line with earlier studies (e.g. Lorenz et al., 2016). The cooling increases towards the end of the 21$^{st}$ century in all three simulations (Figure S3d to f).

### 3.2.2 Prescribing soil ice

In this section we analyse PRES_LIQ+ICE_MEAN and PRES_LIQ+ICE_MEDIAN, i.e. the simulations that prescribe ICE. Using the PRES_LIQ+ICE methodology leads to a similar anomaly in global land mean temperature in the 1971 to 2000 period than prescribing LIQ only (PRES_LIQ+ICE_MEAN: $-0.8$ °C and PRES_LIQ+ICE_MEDIAN: $-0.3$ °C, Figure S4). However, these land temperature differences increase strongly toward the end of the 21$^{st}$ century (Figure S5) in contrast to the simulations without prescribed ICE. As the climate and hence the soils warm, the soil ice melts, and, as the ICE climatology is based on the time period 1971 to 2000, more soil ice is prescribed. Consequently, melting occurs during every modelling time step and the soil ice is re-prescribed at the end of the time step, thereby constantly cooling the land surface and hence near-surface temperature. Thus, prescribing soil ice leads to a strong disturbance of the model's energy balance. This is also evident in the large ground heat flux anomalies of the simulations with prescribed ICE (more than $10$ $\mathrm{Wm}^{-2}$ locally and $1.9$ $\mathrm{Wm}^{-2}$ globally for 2070 to 2099, Figure S6). In contrast, experiments that do not prescribe ICE do not show any noteworthy ground heat flux anomalies. As the climate warms, there is an increasing land area where the air temperature is no longer consistent with a frozen ground, thus the land mean temperature anomaly increases with time. The largest temperature signal occurs locally in the mid- and high latitudes. However, non-local effects due to heat advection and/ or altered atmospheric circulation can not be excluded.

Note that most climate models, for instance within GLACE-CMIP5, do not prescribe ICE and thus do not suffer from this problem. However, ICE was prescribed in CESM in earlier studies (Koster et al., 2004; Lorenz et al., 2012; Seneviratne et al., 2013). This may have caused an increased temperature perturbation that does not affect the main conclusions of these studies. In the GLACE experiments Koster et al. (2004) simulate a summer in the current climate, which reduces the influence of prescribing ICE. Additionally, they concentrate their analysis on the variability of precipitation on non-ice land points. Seneviratne et al. (2013) compare two simulations that both prescribe ICE, such that the effects cancel while others excluded CESM simulations from their analysis (e.g. Berg et al., 2016).

### 3.2.3 Interactive fraction of liquid and frozen soil water

The last two simulations, PRES_FRAC_MEAN and PRES_FRAC_MEDIAN, prescribe total SM, while the relative proportions of LIQ and ICE are interactively computed by the model. Hence, this technique should circumvent the problem of repeatedly adding and melting ICE. However, due to vertical liquid water transport in the soil it also leads to large temperature and ground heat flux anomalies in CLM4 (Figure S4 and Figure S6). In contrast to PRES_LIQ+ICE the annual mean temperature anomaly is already apparent for the period 1971 to 2000 and increases only slightly toward the end of the 21st century (Figure S5). Nonetheless, we think that this technique is viable, and that the problem reported here is CLM4-specific. For example, Figure 2 in Douville et al. (2016) gives no indication of a large temperature anomaly due to the prescription of ICE. It is recommended to calculate the ground heat flux anomalies when prescribing SM, as this is a good indicator of ICE-induced energy balance perturbations.

### 3.3 Amount of prescribed soil moisture

SM is usually prescribed to suppress the land-atmosphere coupling. This comes at the cost of water balance perturbations. To quantify the introduced imbalance, we separately compute the total of (intendedly) added and removed SM for all simulations with respect to REF (for which it is zero). During 1971 to 2000, the average amount of added SM (over the whole soil column) is about $650 \mathrm{~mm~year}^{-1}$ (not shown). This is about three quarters of the global land mean precipitation in REF. However, a similar amount of SM is removed and the net water balance perturbation is much smaller because positive and negative perturbations largely compensate when integrated over the entire soil column. A large amount of water is usually removed from the uppermost soil layers because rain infiltrates the topmost soil layer but has not enough time to reach deeper soil layers before this wet SM is replaced with a (usually) drier climatological value at the end of the time step. Consequently, the deeper layers are too dry and water is added by prescribing the climatological SM.

For these reasons we focus on the net water balance perturbations in the remainder of this Section. In PRES_LIQ_MEAN (Figure 5a), comparatively large amounts of water ($> 250 \mathrm{~mm~year}^{-1}$) are added in Australia, India, Mainland Southeast Asia (Indochina), southern Brazil and parts of Africa. The regions with large amounts of net added SM coincide with regions where we find the strongest TXx reductions in Figure 4, a consequence of the (muted) land-atmosphere coupling. These regions show large positive anomalies in evapotranspiration, which is responsible for the large amounts of added LIQ, as well as the reduction of the sensible heat flux, which in turn leads to lower TXx. Interestingly, TXx decreases almost at all land grid points, while in many regions more water is removed than added. This is explained by evapotranspiration which increases in most land areas (not shown) thus indicating that the SM prescription ensures availability of water even during hot and dry periods. To set the water-balance perturbations into perspective, we scaled the amount of net SM changes by the annual mean precipitation at each grid cell (Figure 5d for PRES_LIQ_MEAN). In many regions, the net water-balance perturbation is more than $30 \%$ of the annual mean precipitation amount (Figure 5d). Not surprisingly, we find the largest relative changes in regions with large absolute SM changes, but also regions with small precipitation amounts (Sahara, Arabian Peninsula).

Simulations with prescribed median SM generally display smaller water-balance perturbations. In PRES_LIQ_MEDIAN (Figure 5b and e), the net water-balance perturbation is generally below $200 \, \text{mm year}^{-1}$. This corresponds to a perturbation of less than $15 \, \%$ of annual mean precipitation in most regions. Regions where less water is added in PRES_LIQ_MEDIAN than PRES_LIQ_MEAN also show substantially smaller evapotranspiration, because the median SM climatology is smaller than the mean. On the other hand, regions where more water is added with the median SM climatology, often show more rainfall, especially northern Brazil. Results for PRES_LIQ_DEEP_MEDIAN (Figure 5c and f) are similar, with the exception that the land area where SM amounts larger than $30 \, \%$ of annual mean precipitation are removed is strongly reduced, probably because water infiltrated in the topmost layer is evaporated (or persists in this layer) instead of removing it by the algorithm.

In terms of global net SM changes, PRES_LIQ_DEEP_MEDIAN introduces the smallest water balance perturbation of all simulations, ($-2 \, \text{mm year}^{-1}$, during 1971 to 2000). This is only slightly more in the case of PRES_LIQ_MEDIAN ($-5 \, \text{mm year}^{-1}$). We find stronger water balance perturbations in PRES_LIQ_MEAN ($43 \, \text{mm year}^{-1}$). Note that in individual years, the water balance perturbations can be larger (Figure 6a). Until the middle of the 21$^{\text{st}}$ century these perturbations are relatively constant for all three simulations and decrease thereafter. Thus, the small negative anomalies in PRES_LIQ_MEDIAN and PRES_LIQ_DEEP_MEDIAN become about $-45 \, \text{mm year}^{-1}$ for 2070 to 2099. For PRES_LIQ_MEAN, on the other hand, the large positive water balance perturbations decrease to $5 \, \text{mm year}^{-1}$. This is caused by increased rainfall over land, which is only partially compensated by increased evapotranspiration (Figure 6b and c). As the mean SM climatology is generally wetter than the median climatology, this wettening brings the interactively computed SM closer to the mean climatology, such that less water balance perturbations are introduced by the SM prescription. Consequently, there is also an increase in global land mean total SM in REF (Figure 6d) in the CESM model. Note that this stands in contrast to other models (Berg et al., 2016), which mostly display drying trends over land. In these models, the water-balance perturbation for prescribing the mean SM climatology would probably increase and not decrease in the future. Thus, on global maps of net water balance perturbations for 2071 to 2100 (Figure S7), the regions with large amounts of added SM are similar as shown in Figure 5a, but more regions show larger amounts of removed SM.

## 4    Conclusions

Soil moisture is commonly prescribed in General Circulation Models to study the interplay of the land surface with weather and climate. As other types of sensitivity experiments (e.g. prescribing sea surface temperatures), this approach introduces perturbations, in particular to the land water balance, because it artificially removes rainwater that infiltrates the soil and replaces water in the soil that is lost via evapotranspiration and drainage. It is important to be aware of these perturbations because they induce changes in the surface climate and constitute a substantial fraction of the climate response to the prescribed soil moisture conditions. Thus, independent experiments investigating the impact of soil moisture-climate interactions may come to different conclusions if they use different approaches to decouple the land surface. However, perturbing the water balance is necessary and cannot be avoided when aiming at an estimation of the land-atmosphere coupling strength. Therefore,

we investigate the impact of different prescription techniques on climate, and, for the first time, also report the water balance perturbations induced by soil moisture prescription.

We implement and test four approaches to prescribe soil moisture, and use two methods to estimate the soil moisture climatology (mean and median) in the Community Earth System Model (CESM) with its land component, the Community

Land Surface Model (CLM). We show that the mean and median soil moisture climatologies differ, with the most notable relative differences in the uppermost soil layers. This difference is also observed in other General Circulation Models within GLACE-CMIP5.

The first method to prescribe soil moisture that was originally developed for CESM/ CLM does not only prescribe soil liquid water but also soil ice (e.g. simulations contributing to GLACE experiments, Koster et al., 2006). This leads to large anomalies

in the ground heat flux and the global mean temperature, especially toward the end of the 21$^{st}$ century, and is therefore generally not recommended. Similar problems are apparent in CLM (version 4) when total soil moisture is prescribed while computing the relative proportions of soil liquid water and soil ice by the model. We propose an alternative methodology where no soil moisture is prescribed if the soil temperature in a particular layer is below freezing point, and only soil liquid water is prescribed otherwise. This method remedies the large global mean temperature and ground heat flux bias of the first method, while it still

allows to mute the land-atmosphere coupling. For this method, we compare the difference between using the mean and the median soil moisture climatology. When prescribing the mean climatology, large net water balance perturbations arise (global land mean of $50$ mm year$^{-1}$, for 1971 to 2000). Whereas in the case of prescribing the median soil moisture climatology, the land mean water balance perturbation is much smaller ($-5$ mm year$^{-1}$). Thus, prescribing the median soil moisture climatology leads to a considerably smaller perturbation of the water balance. However, long-term soil moisture trends may

also influence the water balance perturbations when prescribing a fixed (past) SM climatology. This illustrates the utility of reporting the water balance perturbations, which is also planned within LS3MIP (van den Hurk et al., 2016).

Corresponding to different water balance perturbations, there are different impacts on temperature: when prescribing the mean soil moisture climatology we find a land mean cooling of more than $0.5$ °C, while prescribing the median leads to a mean land cooling of only $0.3$ °C. Regionally, temperature differences of $2$ °C are observed when prescribing the two

climatologies. Our results allow to disentangle the influence of the soil moisture-temperature coupling and the influence of the water-balance perturbation.

For comparison, we furthermore test another well-established method (Koster et al., 2004; Douville, 2003) to prescribe soil moisture where the topmost soil layer is computed interactively and soil moisture is only prescribed in the lower layers. Results with this method are very similar to the findings obtained when prescribing the whole soil column. Due to the interactive top

layer, the water-balance perturbation, and also the temperature signal are slightly smaller.

This study shows that a careful design of the soil moisture prescription methodology can help to minimize its influence on the model climate. Therefore, Table 2 provides a summary of our findings, and recommendations for the set up of studies prescribing soil moisture. These recommendations can guide the implementation of the LFMIP experiments within LS3MIP. Particularly, the method to prescribe soil moisture is not specified within LS3MIP and this paper can serve as reference for

model developers. We note that the originally planned LS3MIP setup mentions the use of the mean climatological soil moisture

(van den Hurk et al., 2016). Using the median climatology as recommended in this study would require a small adaptation of the protocol, but this may still be possible as the simulations are not yet started.

As the land-atmosphere coupling is removed in all experiments in this study, the observed differences in the temperature signals are solely related to differences between the induced water balance perturbations. While these perturbations are inevitable for suppressing the land-atmosphere coupling, our results suggest that the role of these perturbations for the resulting temperature signal is not negligible. Hence, not the entire temperature signal can be attributed to the land-atmosphere coupling. This problem can be addressed by prescribing the median SM climatology, which helps to reduce water balance perturbations because of the non-symmetrically distributed SM in many regions.

## 4.1 Code availability

The used code is available at https://github.com/IACETH/prescribeSM_cesm_1.2.x, where the documentation is linked. The code is released under a MIT license. Revision 67cf64 was used to conduct simulations 1 to 5 and revision c38753 for simulations 6 and 7. Note that the model framework (and code) of CESM/ CLM is necessary to compile and use the code given in the repository.

*Author contributions.* M.H. mainly performed the analysis and wrote the manuscript. All authors participated in the design of the experiments, discussion of the results and writing of the paper.

*Acknowledgements.* This research was funded by the ERC DROUGHT-HEAT project (Contract No. 617518). We thank Ruth Lorenz for discussion of the manuscript and Urs Beyerle for support with CESM. Parts of the employed source code was originally developed by Ruth Lorenz and Dave Lawrence.

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

**Table 1.** Names of simulations used in the study.

| Number | Name | Soil Moisture Climatology |
|---:|---|---|
| 0 | REF | — |
| 1 | PRES_LIQ_MEAN | mean |
| 2 | PRES_LIQ_MEDIAN | median |
| 3 | PRES_LIQ_DEEP_MEDIAN | median |
| 4 | PRES_LIQ+ICE_MEAN | mean |
| 5 | PRES_LIQ+ICE_MEDIAN | median |
| 6 | PRES_FRAC_MEAN | mean |
| 7 | PRES_FRAC_MEDIAN | median |

**Table 2.** Summary of the findings and recommendations for prescribing soil moisture in land surface models.

| | |
|---|---|
| Whole column vs. subsurface prescription of soil moisture | Prescribing soil moisture in subsurface soil levels only, rather than the entire soil column, leads to a marginally smaller water balance perturbation and atmospheric response. |
| Soil moisture climatology (median vs. mean) | Prescribing the median rather than the mean soil moisture leads to a considerably smaller perturbation of the water balance and also of the atmospheric response. |
| Temporal resolution of the soil moisture climatology (daily vs. monthly soil moisture values) | Daily soil moisture follows the seasonal cycle more closely and avoids the difference in monthly means of the reference simulation and the simulation with prescribed soil moisture. While not tested with simulations in this study, the differences in terms of water balance and temperature perturbations when prescribing true daily versus interpolated monthly SM (see Section 3.1.2) may regionally be as large as the ones we find between prescribing mean versus median seasonal SM cycles. |
| Water-balance perturbation as output | We recommend to output the amount of water that is added/ removed by the algorithm as this may help to disentangle the water-balance perturbation and the land-atmosphere coupling. |
| Prescribing soil ice | Prescribing soil ice leads to large temperature and ground heat flux anomalies. To prevent such anomalies the soil moisture prescription should be stopped as soon as the soil reaches freezing temperature. It should thus be ensured that the ice (or water to ice ratio) in the soil can evolve freely. If soil ice should nevertheless be prescribed, using a running median of soil ice and liquid of the control simulation will lead to the smallest perturbations. |

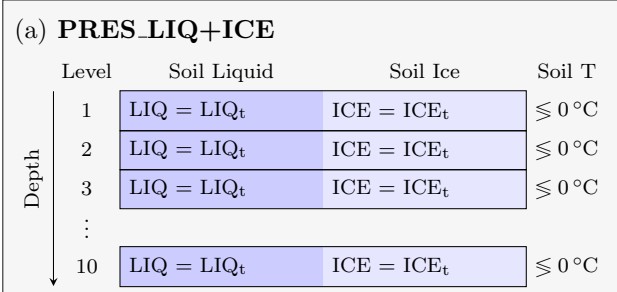

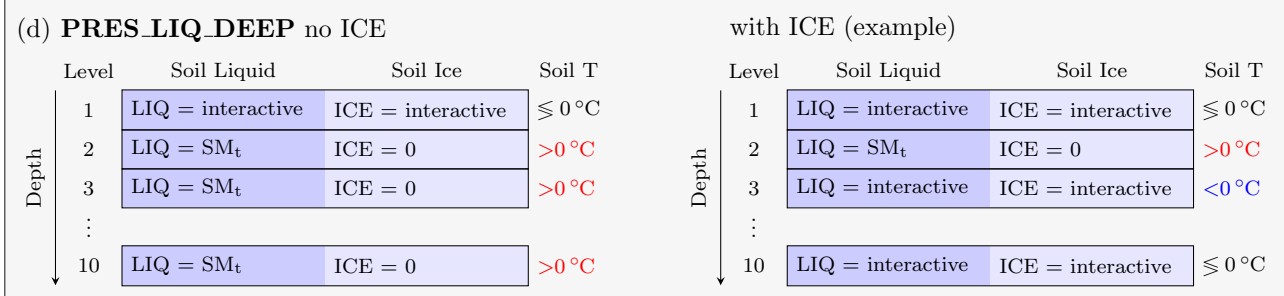

LIQ$_t$, ICE$_t$, and SM$_t$ = LIQ$_t$ + ICE$_t$: target LIQ, ICE, and total SM values
$f$ = LIQ/(LIQ + ICE): fraction of LIQ

**Figure 1.** The four tested approaches to prescribe SM in CLM. The target, LIQ, ICE, and SM values are denoted LIQ$_t$, ICE$_t$, and SM$_t$, respectively. SM$_t$ corresponds to the sum of LIQ$_t$ and ICE$_t$ (i.e. SM$_t$ = LIQ$_t$+ ICE$_t$). In general the target values depend on time (day of year), location (grid point), and depth (soil level). In this study we use the 30-year mean and median seasonal cycle, however, other targets are possible, e.g. a specific year. (a) LIQ and ICE are both prescribed in PRES_LIQ+ICE. (b) In PRES_FRAC, total SM is prescribed, but the fraction, $f$ = LIQ/(LIQ + ICE) is interactively computed by the model. Note that the hydrology in CLM4 is still active. (c) Illustration of the new approach (PRES_LIQ), prescribing LIQ in all soil levels if the soil temperature is above freezing (left) and for an example with soil level two below freezing (right). (d) PRES_LIQ_DEEP: as PRES_LIQ but the first soil layer is always interactive.

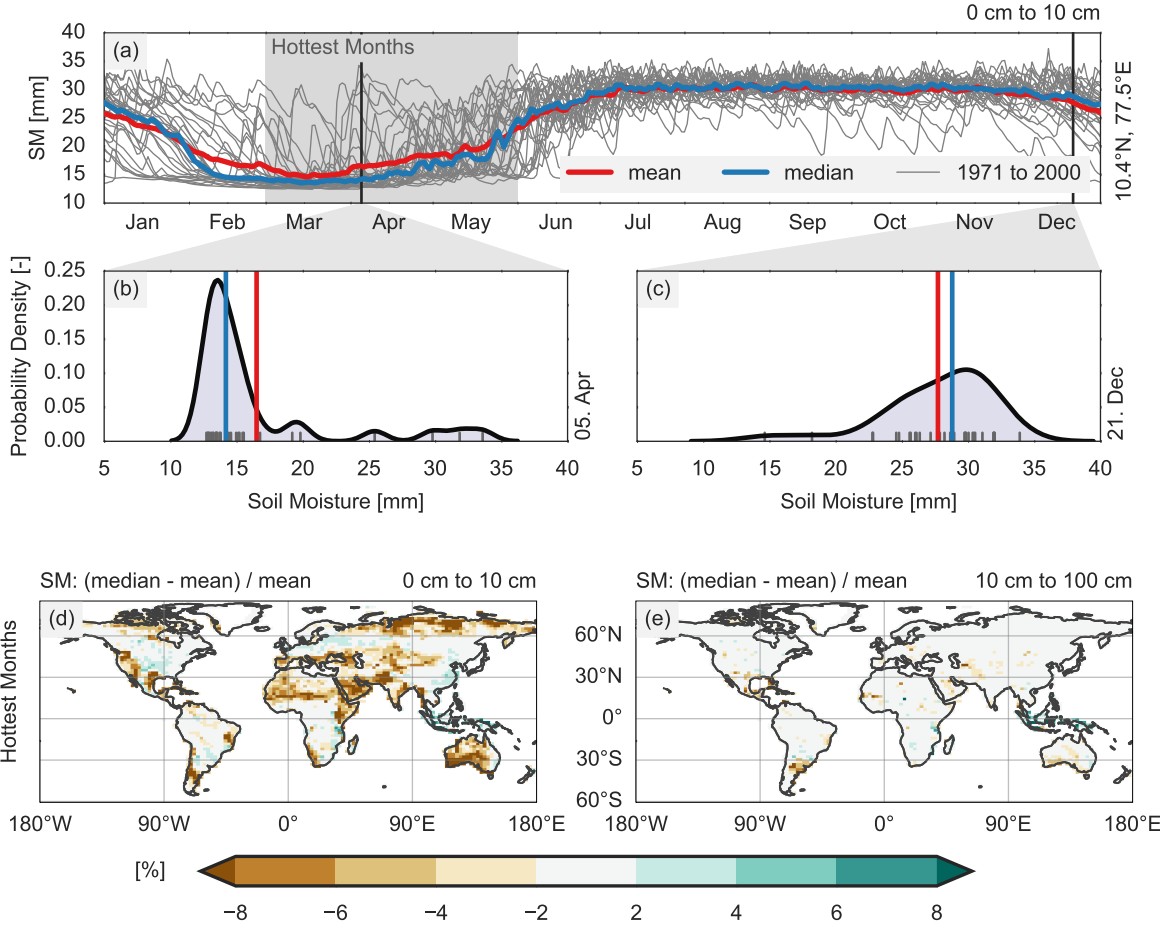

**Figure 2.** Difference between mean and median SM. (a) Seasonal cycle of total SM in the top 10 cm for an example grid point in India (10.4 °N, 77.5 °E) as simulated by CLM for the climatological period (1971 to 2000). Shown are the individual years (gray lines), and their mean (red) and median (blue). Light gray background shows the three consecutive hottest months at this grid point and vertical black lines the two days depicted in (b) and (c), respectively. (b) and (c) Kernel density estimate of the SM distribution (thick black line), including the individual years (thin gray lines) and the mean (red) and median (blue) SM values for the 5[th] of April (b) and 21[st] of December (c). (d) and (e) Relative difference in the SM climatology between median and mean for the hottest months of the year in the surface layer (0 cm to 10 cm, d) and in 10 cm to 100 cm depth (e).

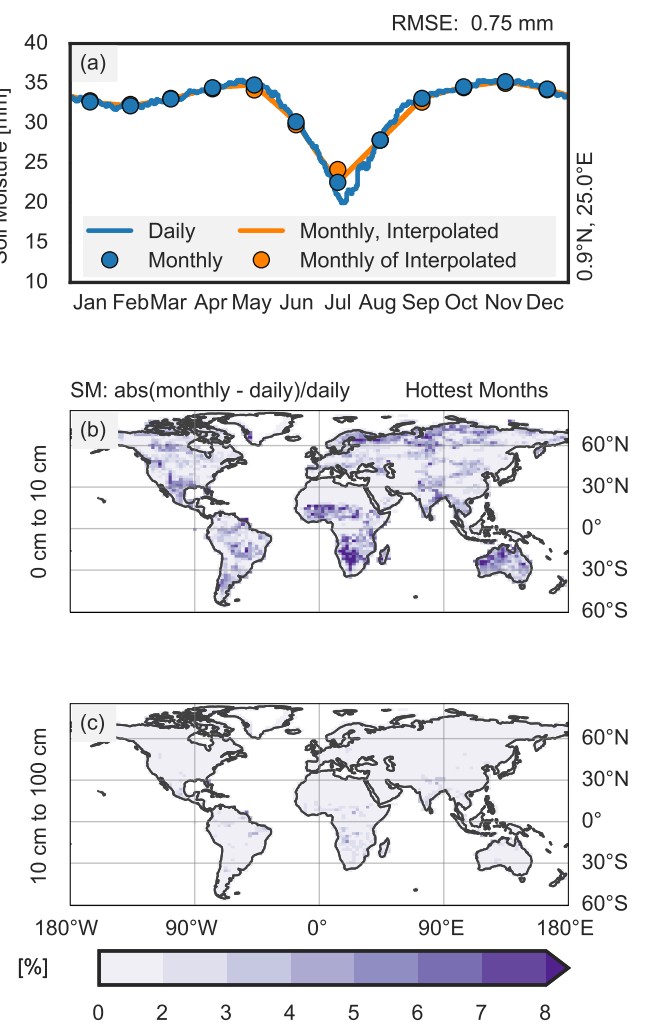

**Figure 3.** Difference between interpolated monthly and daily SM. (a) Seasonal cycle of median SM climatology for one grid point in Central Africa (0.9 °N, 25 °E), illustrating the difference between daily and interpolated monthly values. (b) and (c) Absolute difference [%] in the median SM climatology between daily and interpolated monthly values in the surface layer (0 cm to 10 cm, b) and in 10 cm to 100 cm depth (c).

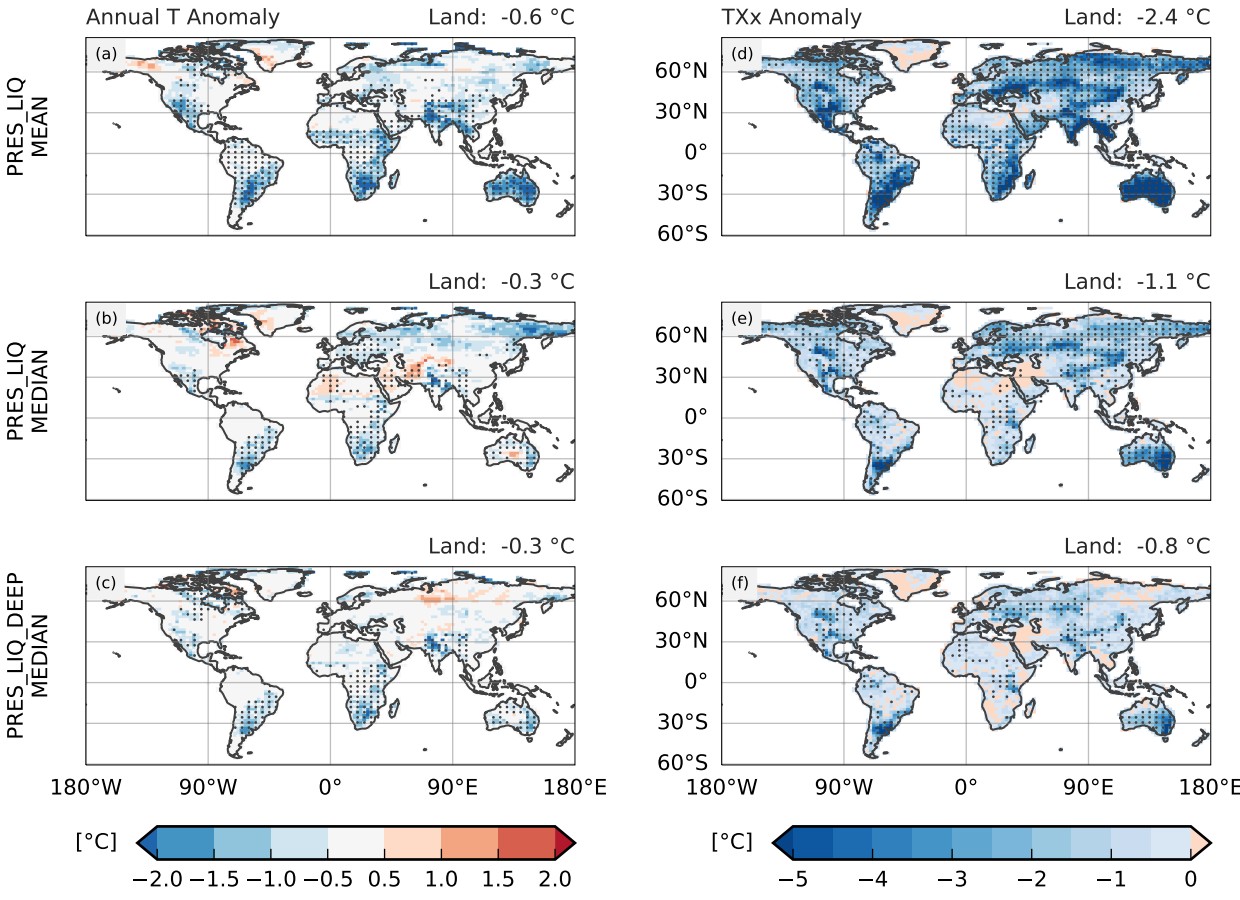

**Figure 4.** Difference in the median of the simulation with prescribed SM and REF (anomaly) for the period 1971 to 2000. (a) to (c) annual mean Temperature, (d) to (f) TXx. Significance is tested with a Wilcoxon-Mann-Whitney-U test (e.g. Wilks, 2011). Conducting a significance tests at each grid point increases the probability to falsely reject the null hypothesis (e.g. Wilks, 2016). We therefore control for this with the approach described by Benjamini and Hochberg (1995), using a global p-value of $5\%$.

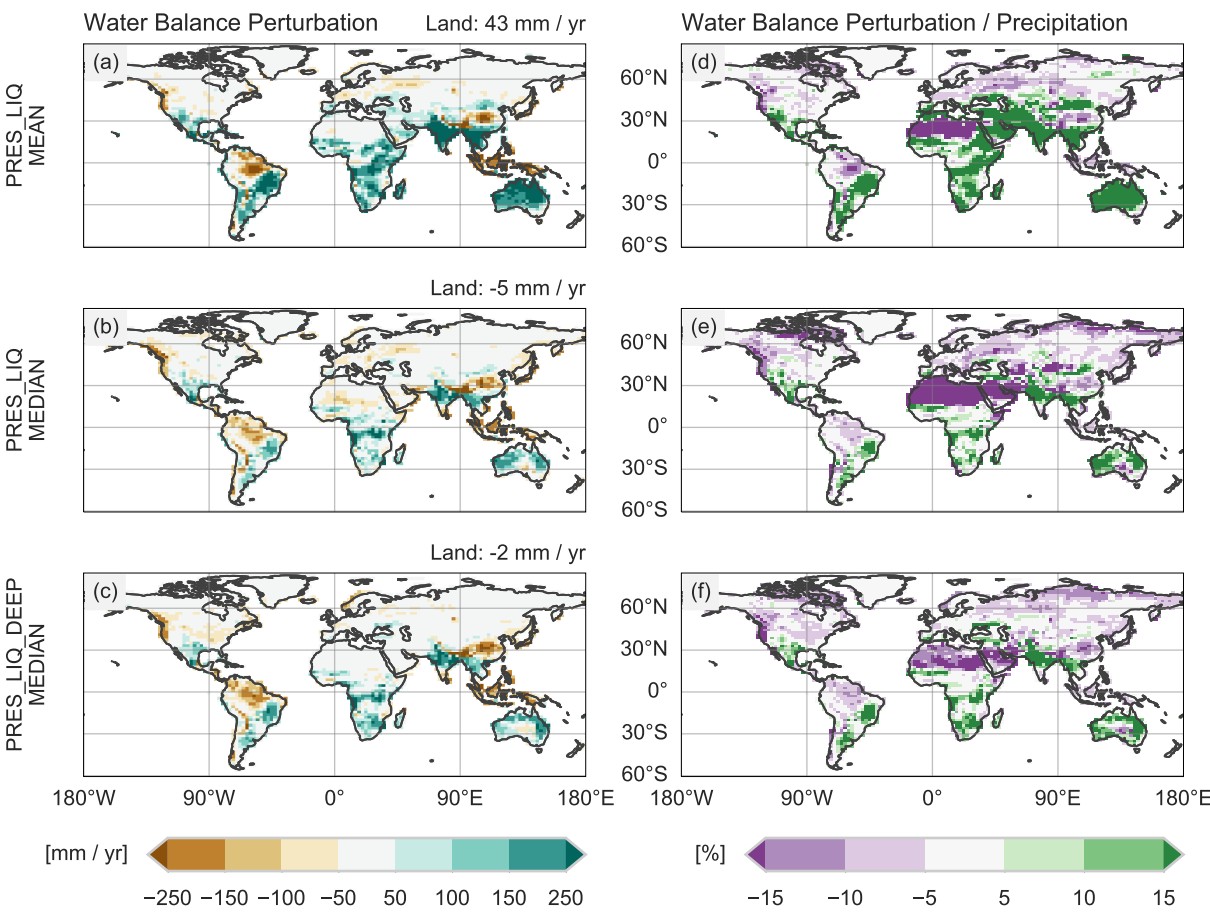

**Figure 5.** Mean annual SM perturbation for 1971 to 2000. (a) to (c) net water balance perturbation in $\mathrm{mm\ year}^{-1}$, (d) to (f) net water balance perturbation scaled by the annual mean precipitation.

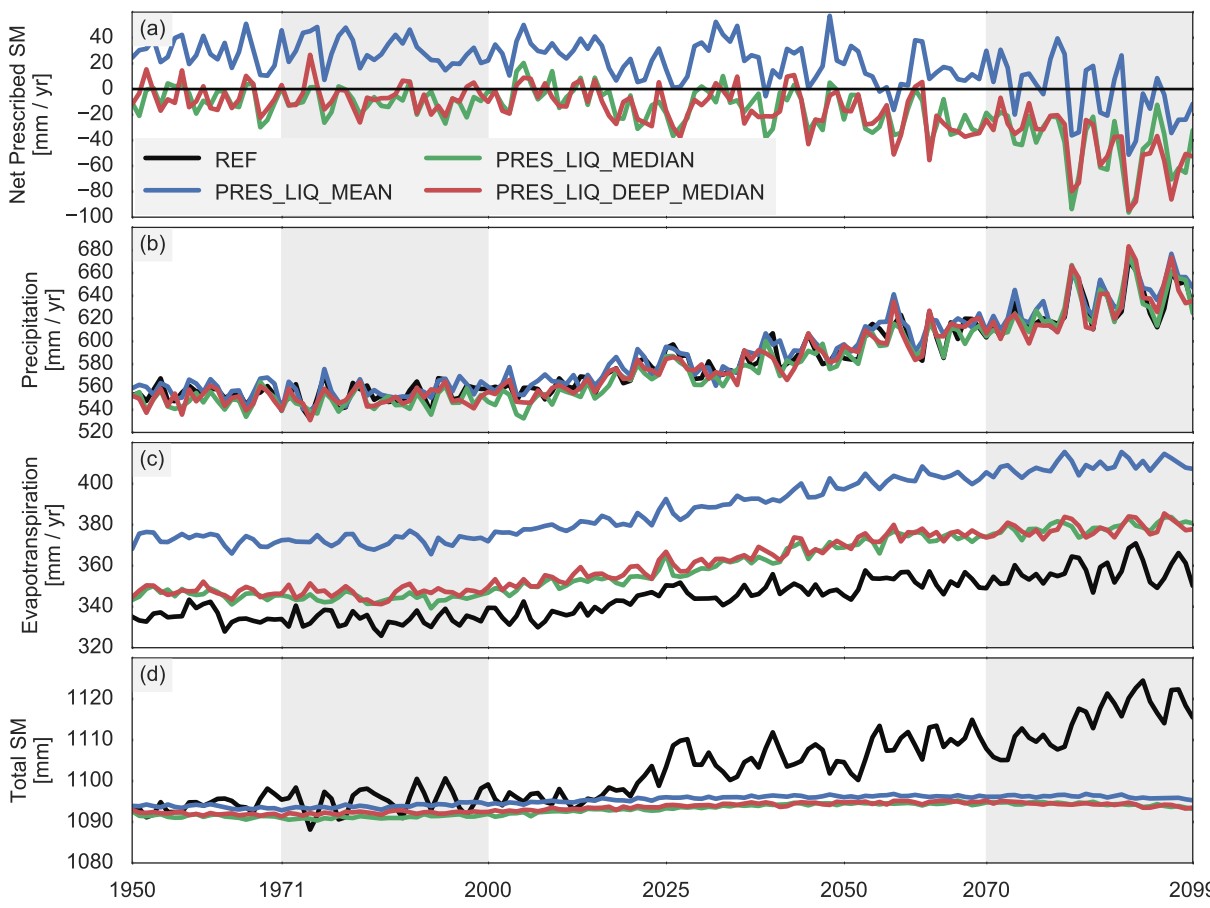

**Figure 6.** Time series of global-land, annual-mean (a) net prescribed soil moisture, (b) precipitation, (c) evapotranspiration and, (d) total soil moisture content. Total soil moisture in the simulations with prescribed soil moisture is not entirely constant because ICE is still computed interactively. The light gray background shows the two time periods used for the climatology.