# Peer review of "Investigating soil moisture-climate interactions with prescribed soil moisture experiments: an assessment with the Community Earth System Model (version 1.2)"

_Geoscientific Model Development, 2016_

## Referee Comment (RC1) · B. van den Hurk (Referee) · 1 Nov 2016

General remarks

This manuscript carries out a timely analysis of the consequences of perturbing the land surface soil moisture budget as carried out in earlier experiments and proposed in LS3MIP. It compares various methodologies (with/without ice, with/without prescribing shallow top layer, using mean/median), concluding that the use of the median liquid is a more conservative method than when using means and including ice. It is well written and addresses an outstanding issue, and is thus worth publishing subject to

some minor comments:

- P4, L4-L8: the algorithm uses a soil temperature threshold of zero degrees to trace the occurrence of soil ice. However, algorithms exist that allow a gradual fraction of soil water to be frozen in between a temperature range that may well include temperatures exceeding 0 degrees. How to deal with these parameterizations?

- P4, L23: when is there "too much variability"?

- P5, L18: it may be worth spending a few words explaining (or speculating) why the soil moisture distribution shows a negative skewness and a median lower than a mean. Is it because soil moisture is more persistent in drier conditions due to lower values of hydraulic exchange coefficients? Or is there another reason behind this assymetry?

- P5, L28: it's not the strength of the seasonality that is at play hear, but the occurrence of a short sharp peak in that climatology, that causes these rounding errors

- P6, L16: suggest to add "when comparing the median to the mean" at the end of this sentence

- P6, L17: the fact that the results in 2070-2099 are similar is surprising. You are not comparing the REF temperature in 1970-1999 to the simulated temperature by the end of the century I presume (otherwise we should have seen a major climate change signal). But also the GLACE-CMIP5 exp by Seneviratne et al (2013) did show an effect on net warming when prescribing a climatological soil moisture. Why is this effect gone in this set-up?

[Figure]

- P7, L11: Koster et al (2004, 2006) did evaluate all perturbations under present climate conditions, which makes the effect of changing frozen soil water also smaller than in climate change set-ups

- P7, L18: I misread this sentence a few times. I would make the statement of 650 mm/yr for the addition of SM first, and then state that a similar amount is associated with removals of soil water. Now it looks like 650 mm/yr is the net effect

- Figures: they are generally pretty small, and stippling is difficult to see

---

## Referee Comment (RC2) · J. Colin (Referee) · 4 Jan 2017

General comments

This paper investigates some of the issues related to the experimental protocol of the "Land Surface, Snow and Soil Moisture Model Intercomparison Project" (LS3MIP). Several methods to prescribe the soil moisture conditions are tested, and the results are analyzed in terms of water balance perturbations. This constitute a new diagnostic that should be quite inspiring for other modelling groups. The study is carefully carried out and well written. And it is highly relevant in the context of the coming LS3MIP

exercise. I recommend a publication, although I have some minor comments.

Specific comments

1. p. 4, lines 1-15 (description of the various methods of prescription)

The authors consider the possibility of prescribing either the liquid water content only, or the liquid and ice contents separately. There is (at least) another option in which the total amount of soil moisture (liquid + ice) is prescribed and the partition of ice and water is computed accordingly to the model's proportion of liquid and ice at a given time step (i.e. before the value is prescribed). This what we did in Douville et al. (2016) and we tend to think this method can prevent most of the disturbance in the energy balance you observe with the PRES_LIQ+ICE method. It would have been interesting to test it. But since it was not, it could be worth mentioning.

2. p. 4, lines 4-8

I had a hard time understanding the description of the PRES_LIQ method. Figure 1 definitely clarifies things, but the written explanations should be improved. For example, the text could explicitly mention that the total soil moisture content is converted into liquid water to be prescribed. The authors could also write that below zero, both the liquid water and ice contents are let interactive.

3. p.7, lines 26-27

"Interestingly, the regions with large amounts of net added SM coincide with regions where we find the strongest Txx reduction in Figure 4". Could you give some physical explanations of this finding?

4. p.7, lines 26-27

The reduction of TXx found in southwestern Europe in figure 4.d does not match any perturbation of water balance in figure 5.d. Can you comment on that?

5. p. 7 line 32 to p.8 line 4

Do you have some insights as to why the PRES_LIQ_MEDIAN method leads to a smaller imbalance than the PRES_LIQ_MEAN one? It would help to plot the distribution function of SM, as in figure 2.a, for grid points where the differences between the two methods are the greatest. Let's say in India where large amounts of water are added in PRES_LIQ_MEAN and in Indonesia or Brazil where water is removed.

Technical corrections

1. The figures should be enlarged.
* * *

---

## Author Response (AR1)

**Response to Reviewers for "Investigating soil moisture-climate interactions with prescribed soil moisture experiments: an assessment with the Community Earth System Model (version 1.2)"**

Mathias Hauser[1], René Orth[1] and Sonia I. Seneviratne[1]
[1]Institute for Atmospheric and Climate Science, ETH Zurich, Zurich, Switzerland

We are thankful to the reviewers for their positive comments and their feedback, which helped us to improve the manuscript. We added the following main changes to the revised manuscript:

- Simulations with a new methodology to prescribe soil moisture, and its discussion. The new methodology prescribes soil water and ice but lets the model determine the relative proportions of the two components (PRES_FRAC).
- A more thorough discussion of the skewed soil moisture distribution, and the temperature response to the soil moisture prescription
- All figures were enlarged. Some figures were updated to include the new prescription method (Figure 1, Figure S4, and Figure S5). Figure 2 was enhanced to include time series of soil moisture for a whole year for an example grid point. Figure S9 was moved to the main text and is now Figure 6. Figure S6 and Figure S7 were removed to reduce the number of figures in the supplementary material. We added a new Figure S6 to show the ground heat flux anomalies for all seven simulations.
- Some minor adaptations to the manuscript text.

**Reviewer 1 (Bart van den Hurk)**

General remarks
This manuscript carries out a timely analysis of the consequences of perturbing the land surface soil moisture budget as carried out in earlier experiments and proposed in LS3MIP. It compares various methodologies (with/without ice, with/without prescribing shallow top layer, using mean/median), concluding that the use of the median liquid is a more conservative method than when using means and including ice. It is well written and addresses an outstanding issue, and is thus worth publishing subject to some minor comments.

**A1:** We thank the reviewer for the encouraging comments.

P4, L4-L8: the algorithm uses a soil temperature threshold of zero degrees to trace the occurrence of soil ice. However, algorithms exist that allow a gradual fraction of soil water to be frozen in between a temperature range that may well include temperatures exceeding 0 degrees. How to deal with these parameterizations?

**A2:** The important part of the new algorithm is to not artificially add (or remove) soil ice. We recommend stopping the prescription as soon as ice appears in the soil, even if the temperature is larger than 0° C. We added this point to the manuscript on P4 L8:
The important characteristic of this new algorithm is that it never artificially adds ICE (see Section 3.2.2). Although (supercooled) LIQ and ICE can coexist in CLM4, we leave the soil hydrology entirely interactive below the freezing temperature.

And in Table 2 (Prescribing soil ice):
To prevent such anomalies the soil moisture prescription should be stopped as soon as the soil reaches freezing temperature.

P4, L23: when is there "too much variability"?

**A3:** We have not considered sub-daily soil moisture variations (or the effect thereof) and we have rephrased this paragraph to reflect this on P4 L26:
In this study we use daily mean values as linearly-interpolated monthly values can be too coarse (see below).

P5, L18: it may be worth spending a few words explaining (or speculating) why the soil moisture distribution shows a negative skewness and a median lower than a mean. Is it because soil moisture is more persistent in drier conditions due to lower values of hydraulic exchange coefficients? Or is there another reason behind this asymmetry?

**A4:** We added a new paragraph discussing the skewed distribution of SM on P5 L15:
In the dry season the median is generally smaller than the mean, with large rainfall events leading to outliers on the wet end of the distribution. For example on the 5$^{th}$ of April (Figure 2b), the difference is -2.3 mm, or -14.0%. During the wet period the median is usually larger than the mean, here it is dry years that lead to the asymmetry. However, the difference between median and mean are generally smaller, on the 21$^{st}$ of December, for example, (Figure 2c) it is 1.0 mm, or 3.8%. There are many processes that contribute to non-symmetric SM distributions: the positive skewed distribution of precipitation, the upper and lower bound in the water holding capacity of the soil (between the wilting point and saturation), as well as the strong nonlinear function of water flow (hydraulic conductivity) within the soil with respect to the SM state (Laio et al., 2001).

P5, L28: it's not the strength of the seasonality that is at play hear, but the occurrence of a short sharp peak in that climatology, that causes these rounding errors

**A5:** We rewrote the sentence on P6, L9:
True daily and interpolated monthly SM values can differ in regions with a short sharp peak in the seasonal cycle, as exemplified for a grid point in Central Africa (Figure 3a).

P6, L16: suggest to add "when comparing the median to the mean" at the end of this sentence.

**A6:** We added this to the sentence on P6, L25:
PRES_LIQ_MEDIAN has smaller temperature anomalies than RES_LIQ_MEAN, corresponding to the regions with smaller climatological SM when comparing the median to the mean (Figure 2).

P6, L17: the fact that the results in 2070-2099 are similar is surprising. You are not comparing the REF temperature in 1970-1999 to the simulated temperature by the end of the century I presume (otherwise we should have seen a major climate change signal). But also the GLACE-CMIP5 exp by Seneviratne et al (2013) did show an effect on net warming when prescribing climatological soil moisture. Why is this effect gone in this set-up?

**A7:** It is correct that we compare EXP – REF for 2070 to 2099 (where EXP is one of the experiments with prescribed SM).

The largest part of the global mean temperature signal is lost by not prescribing ICE. A second reason comes from taking the median in time instead of the mean (as in Seneviratne et al., 2013). For

example for "PRES_LIQ_MEAN – REF", the median for "2070 to 2099" minus "1971 to 2000" is -0.04° C (which is lost in the Figure title due to truncation), while for the mean it is -0.16 °C. In GLACE-CMIP5 the difference in warming for "EXPA – CTL" for the global land is -0.38 °C for the multi model mean, and -0.81 °C, -0.35 °C, -0.16 °C, -0.34 °C, -0.25 °C for CESM, EC-EARTH, ECHAM, GFDL, and IPSL. Thus, PRES_LIQ_MEAN is in the range of GLACE-CMIP5 models.

We addressed both points on P6, L27:
We find similar results when comparing the experiments to REF for the time period 2070 to 2099 (Figure S3 a to c). Thus, the global land warming between 1971 to 2000 and 2070 to 2099 is only slightly larger in REF than the experiments. This is in line with earlier findings (Seneviratne et al., 2013), although experiments in this study are at the lower end of the range of GLACE-CMIP5 models.

P7, L11: Koster et al (2004, 2006) did evaluate all perturbations under present climate conditions, which makes the effect of changing frozen soil water also smaller than in climate change set-ups.

**A8:** This is correct; we included this information on P7, L24:
In the GLACE experiments Koster et al. (2004) simulate a summer in the current climate, which reduces the influence of prescribing ICE.

P7, L18: I misread this sentence a few times. I would make the statement of 650 mm/yr for the addition of SM first, and then state that a similar amount is associated with removals of soil water. Now it looks like 650 mm/yr is the net effect.

**A9:** We rewrote the sentence as suggested on P8, L8:
During 1971 to 2000, the average amount of added SM (over the whole soil column) is about 650 mm year-1 (not shown). This is about three quarters of the global land mean precipitation in REF. However, a similar amount of SM is removed and the net water balance perturbation is much smaller because positive and negative perturbations largely compensate when integrated over the entire soil column.

Figures: they are generally pretty small, and stippling is difficult to see.

**A10:** We updated the figures.

**Reviewer 2 (Jeanne Colin)**

General comments
This paper investigates some of the issues related to the experimental protocol of the "Land Surface, Snow and Soil Moisture Model Intercomparison Project" (LS3MIP). Several methods to prescribe the soil moisture conditions are tested, and the results are analyzed in terms of water balance perturbations. This constitute a new diagnostic that should be quite inspiring for other modelling groups. The study is carefully carried out and well written. And it is highly relevant in the context of the coming LS3MIP exercise. I recommend a publication, although I have some minor comments.

**B1:** We thank the Jeanne Colin for these positive comments.

Specific comments
1. p. 4, lines 1-15 (description of the various methods of prescription) The authors consider the possibility of prescribing either the liquid water content only, or the liquid and ice contents separately. There is (at least) another option in which the total amount of soil moisture (liquid + ice) is prescribed and the partition of ice and water is computed accordingly to the model's proportion of liquid and ice at a given time step (i.e. before the value is prescribed). This what we did in Douville et al. (2016) and we tend to think this method can prevent most of the disturbance in the energy balance you observe with the PRES_LIQ+ICE method. It would have been interesting to test it. But since it was not, it could be worth mentioning.

**B2:** We also performed such simulations with CESM/ CLM, and we added them to the paper as PRES_FRAC_MEAN and PRES_FRAC_MEDIAN. Unfortunately, this also led to large temperature/ ground heat flux anomalies in CLM4. We suspect that vertical liquid water transport in the soil is responsible for this (when soil ice melts the water ends up in a different soil layer than where it originates from and the fraction of soil ice is still 100 %, thus soil ice is added). However, we do think that this technique is valuable, and that this is a CLM4-specific problem.
Given Figure 2 in Douville et al., 2016 (especially "FR – FNF", and "PNP – PR"), we are confident that your simulations do not suffer from this problem.

2. p. 4, lines 4-8
I had a hard time understanding the description of the PRES_LIQ method. Figure 1 definitely clarifies things, but the written explanations should be improved. For example, the text could explicitly mention that the total soil moisture content is converted

into liquid water to be prescribed. The authors could also write that below zero, both the liquid water and ice contents are let interactive.

**B3:** We rewrote the description as suggested on P4, L6:
Furthermore, we propose an alternative approach where SM is only prescribed when the soil temperature is above 0° C (PRES_LIQ). If the soil is frozen, LIQ and ICE are both computed interactively. The climatological total SM (i.e. LIQ + ICE) is converted into LIQ for the prescription.

3. p.7, lines 26-27
"Interestingly, the regions with large amounts of net added SM coincide with regions where we find the strongest Txx reduction in Figure 4". Could you give some physical explanations of this finding?

**B4:** Please see answer B5.

4. p.7, lines 26-27
The reduction of TXx found in southwestern Europe in figure 4.d does not match any perturbation of water balance in figure 5.d. Can you comment on that?

**B5:** We added a paragraph addressing this and the last comment on P8, L17:
The regions with large amounts of net added SM coincide with regions where we find the strongest TXx reductions in Figure 4, a consequence of the (muted) land-atmosphere coupling. These regions also show large positive anomalies in evapotranspiration, which is responsible for the large amounts of added LIQ, as well as the reduction of the sensible heat flux, which in turn leads to lower TXx. Interestingly, TXx decreases almost at all land grid points, while in many regions more water is removed than added. This is explained by evapotranspiration which increases in most land areas (not shown) thus indicating that the SM prescription ensures availability of water even during hot and dry periods.

5. p. 7 line 32 to p.8 line 4
Do you have some insights as to why the PRES_LIQ_MEDIAN method leads to a smaller imbalance than the PRES_LIQ_MEAN one? It would help to plot the distribution function of SM, as in figure 2.a, for grid points where the differences between the two methods are the greatest. Let's say in India where large amounts of water are added in PRES_LIQ_MEAN and in Indonesia or Brazil where water is removed.

**B6:** We added a short discussion in the paper on P8, L29:

[revised manuscript text omitted]

**(a) PRES_LIQ+ICE**

| Level | Soil Liquid | Soil Ice | Soil T |
|---|---|---|---|
| 1 | LIQ = LIQ$_t$ | ICE = ICE$_t$ | $\leqq 0\,°C$ |
| 2 | LIQ = LIQ$_t$ | ICE = ICE$_t$ | $\leqq 0\,°C$ |
| 3 | LIQ = LIQ$_t$ | ICE = ICE$_t$ | $\leqq 0\,°C$ |
| ⋮ | | | |
| 10 | LIQ = LIQ$_t$ | ICE = ICE$_t$ | $\leqq 0\,°C$ |

**(b) PRES_FRAC**

| Level | Soil Liquid | Soil Ice | Soil T |
|---|---|---|---|
| 1 | LIQ = $f\cdot$SM$_t$ | ICE = $(1-f)\cdot$SM$_t$ | $\leqq 0\,°C$ |
| 2 | LIQ = $f\cdot$SM$_t$ | ICE = $(1-f)\cdot$SM$_t$ | $\leqq 0\,°C$ |
| 3 | LIQ = $f\cdot$SM$_t$ | ICE = $(1-f)\cdot$SM$_t$ | $\leqq 0\,°C$ |
| ⋮ | | | |
| 10 | LIQ = $f\cdot$SM$_t$ | ICE = $(1-f)\cdot$SM$_t$ | $\leqq 0\,°C$ |

[Figure]

**(c) PRES_LIQ** no ICE

| Level | Soil Liquid | Soil Ice | Soil T |
|---|---|---|---|
| 1 | LIQ = SM$_t$ | ICE = 0 | $>0\,°C$ |
| 2 | LIQ = SM$_t$ | ICE = 0 | $>0\,°C$ |
| 3 | LIQ = SM$_t$ | ICE = 0 | $>0\,°C$ |
| ⋮ | | | |
| 10 | LIQ = SM$_t$ | ICE = 0 | $>0\,°C$ |

with ICE (example)

| Level | Soil Liquid | Soil Ice | Soil T |
|---|---|---|---|
| 1 | LIQ = SM$_t$ | ICE = 0 | $>0\,°C$ |
| 2 | LIQ = interactive | ICE = interactive | $<0\,°C$ |
| 3 | LIQ = interactive | ICE = interactive | $\leqq 0\,°C$ |
| ⋮ | | | |
| 10 | LIQ = interactive | ICE = interactive | $\leqq 0\,°C$ |

**(d) PRES_LIQ_DEEP** no ICE

| Level | Soil Liquid | Soil Ice | Soil T |
|---|---|---|---|
| 1 | LIQ = interactive | ICE = interactive | $\leqq 0\,°C$ |
| 2 | LIQ = SM$_t$ | ICE = 0 | $>0\,°C$ |
| 3 | LIQ = SM$_t$ | ICE = 0 | $>0\,°C$ |
| ⋮ | | | |
| 10 | LIQ = SM$_t$ | ICE = 0 | $>0\,°C$ |

with ICE (example)

| Level | Soil Liquid | Soil Ice | Soil T |
|---|---|---|---|
| 1 | LIQ = interactive | ICE = interactive | $\leqq 0\,°C$ |
| 2 | LIQ = SM$_t$ | ICE = 0 | $>0\,°C$ |
| 3 | LIQ = interactive | ICE = interactive | $<0\,°C$ |
| ⋮ | | | |
| 10 | LIQ = interactive | ICE = interactive | $\leqq 0\,°C$ |

LIQ$_t$, ICE$_t$, and SM$_t$ = LIQ$_t$ + ICE$_t$: target LIQ, ICE, and total SM values
$f$ = LIQ/(LIQ + ICE): fraction of LIQ

**Figure 1.** The  four tested approaches to prescribe SM in CLM. The target, LIQ, ICE, and SM values are denoted LIQ$_t$, ICE$_t$, and SM$_t$, respectively. SM$_t$ corresponds to the sum of LIQ$_t$ and ICE$_t$ (i.e. SM$_t$ = LIQ$_t$ + ICE$_t$). In general the target values depend on time (day of year), location (grid point), and depth (soil level). In this study we use a 30-year mean or median seasonal cycle, however, other targets are possible, e.g. a specific year. (a) LIQ and ICE are both prescribed in PRES_LIQ+ICE. (b) In PRES_FRAC, total SM is prescribed, but the fraction, $f$ = LIQ/(LIQ + ICE) is interactively computed by the model. Note that the hydrology in CLM4 is still active. (c) Illustration of the new approach (PRES_LIQ), prescribing LIQ in all soil levels if the soil temperature is above freezing (left) and for an example with soil level two below freezing (right). (d) PRES_LIQ_DEEP: as PRES_LIQ but the first soil layer is always interactive.

[Figure]

**Figure 2.** (a) Evolution of total SM for an example grid point in India (10.4 °N, 77.5 °E) as simulated by CLM for the climatological period (1971 to 2000). Shown are the individual years (gray lines), and their mean (red) and median (blue). Light gray background shows the three consecutive hottest months at this grid point and vertical black lines the two days depicted in (b) and (c), respectively. (b) and (c) Kernel density estimate of the SM distribution (thick black line), including the individual years (thin gray lines) and the mean (red) and median (blue) SM values for the 5th of April (b) and 21st of December (c). (d) and (e) Relative difference in the SM climatology between median and mean for the hottest months of the year in the surface layer (0 cm to 10 cm, d) and in 10 cm to 100 cm depth (e).

[Figure]

**Figure 3.** (a) Annual cycle of median SM climatology for one grid point in Central Africa (0.9 °N, 25 °E), illustrating the difference between daily and interpolated monthly values. (b,) and (c) Absolute difference [%] in the median SM climatology between daily and interpolated monthly values in the surface layer (0 0 cm to 10 10 cm, b) and in 10 10 cm to 100 100 cm depth (c).

[Figure]

**Figure 4.** Difference in the median of the simulation with prescribed SM and REF (anomaly) for the period 1971 to 2000. (a) to (c) annual mean Temperature, (d) to (f) TXx.  Significance is tested with a Wilcoxon-Mann-Whitney-U test (e.g. Wilks, 2011). Conducting a significance  tests at each grid point increases the  probability to falsely reject the null hypothesis (e.g. Wilks, 2016). We therefore control for this with the approach described by Benjamini and Hochberg (1995), using a global p-value of 5 %.

[Figure]

**Figure 5.** Mean annual SM perturbation for 1971 to 2000. (a) to (c) net water balance perturbation in $\mathrm{mm\,year^{-1}}$, (d) to (f) net water balance perturbation scaled by the annual mean precipitation.

[Figure]

**Figure 6.** Time series of global-land, annual-mean (a) net prescribed soil moisture, (b) precipitation, (c) evapotranspiration and, (d) total soil moisture content. Total soil moisture in the simulations with prescribed soil moisture is not a totally straight line because ICE is still computed interactively. The light gray background shows the two time periods used for the climatology.

---

## Author Response (AR3)

Dear Editor,

Thank you for the feedback on the manuscript. We updated the text to highlight the contributions of this work to LS3MIP.

The method to prescribe soil moisture is not defined in the LS3MIP experiment setup. Our study presents and compares implementations as a reference for model developers taking part in the experiments. On the other hand, LS3MIP proposes to prescribe the mean climatology while we argue for the median. If this can and should be changed needs to be discussed in the LS3MIP steering committee and our paper provides the basis for this discussion.

We have also made some minor changes to the text to make it more precise.

On behalf of the authors,
Mathias Hauser

[revised manuscript text omitted]